# How Has COVID-19 Impacted Our Language Use?

**DOI:** 10.3390/ijerph192113836

**Published:** 2022-10-24

**Authors:** Francesca Pisano, Alessio Manfredini, Daniela Brachi, Luana Landi, Lucia Sorrentino, Marianna Bottone, Chiara Incoccia, Paola Marangolo

**Affiliations:** 1Department of Humanities Studies, University Federico II, 80133 Naples, Italy; 2IRCCS Santa Lucia Foundation, 00179 Rome, Italy

**Keywords:** social media, misinformation, COVID-19 emergency, metaphors, semantic priming, social communication, psychological disorders, public health

## Abstract

The COVID-19 pandemic has led to severe consequences for people’s mental health. The pandemic has also influenced our language use, shaping our word formation habits. The overuse of new metaphorical meanings has received particular attention from the media. Here, we wanted to investigate whether these metaphors have led to the formation of new semantic associations in memory. A sample of 120 university students was asked to decide whether a target word was or was not related to a prime stimulus. Responses for pandemic pairs in which the target referred to the newly acquired metaphorical meaning of the prime (i.e., “trench”—“hospital”) were compared to pre-existing semantically related pairs (i.e., “trench”—“soldier”) and neutral pairs (i.e., “trench”—“response”). Results revealed greater accuracy and faster response times for pandemic pairs than for semantic pairs and for semantic pairs compared to neutral ones. These findings suggest that the newly learned pandemic associations have created stronger semantic links in our memory compared to the pre-existing ones. Thus, this work confirms the adaptive nature of human language, and it underlines how the overuse of metaphors evoking dramatic images has been, in part, responsible for many psychological disorders still reported among people nowadays.

## 1. Introduction

The coronavirus disease 2019 (COVID-19), caused by the severe acute respiratory syndrome coronavirus 2 (SARS-CoV-2), has been a global epidemic that poses a serious threat to public health throughout the world [1,2]. It has changed the fabric of society worldwide through social distancing measures, travel bans, self-quarantine, and business closures, leading to severe physical and psychological consequences for people’s health [3,4,5,6,7]. Indeed, higher levels of anxiety, depression, and stress have been recorded during the pandemic compared to the pre-COVID-19 emergency disrupting the balance of daily activities and the perception of well-being in healthy [8,9,10,11,12] and neurological populations [13,14,15]. Since the beginning of the pandemic, social media have been primarily used to disseminate crucial information to the public, influencing people’s risk perceptions, decision-making capacity, and behaviors (for a review, see [16]). Several studies have described the COVID-19 times as “an era of fake news” in which misinformation has spread rapidly, in some cases threatening the individual’s mental health and well-being [17,18,19,20,21,22]. Thus, during this period, social media have played a major role in everyday communication; new words and figurative expressions, as well as shifts in the use and meaning of typically sociological, psychological, or even medical terms, have been adopted [23,24,25,26]. Indeed, human language is not a static concept but a dynamic, creative, and adaptive entity with words changing over time due to the actual social context in which interpersonal and social phenomena are incorporated [23,24,27]. Among the figurative expressions, the usage of metaphors has received special attention [25,28,29,30,31]. As reported in the literature, a metaphor is a figure of speech in which a word or phrase literally denoting one kind of object or idea (i.e., trench) is used in place of another (i.e., hospital) to suggest a likeness or analogy between them. Thus, its usage might result in pervasive influencing on the way in which people think and act [32].

Since the beginning of the pandemic, different metaphors have been mostly used to get the public’s attention to the dangers of the virus. Among these expressions, for an emergency, military metaphors such as “war”, “battle”, “fight”, “invasion”, “heroes”, and “warriors” have been employed; while to evoke mortality, “peak”, “wave”, and “spiral” were the most frequent designed metaphors [33]. Thus, following this social media propaganda, these metaphors have become familiar, and they are still widely used in everyday language around the world [24].

Given the emergence of this new vocabulary, in the present study, we wanted to investigate whether the overuse of these metaphors has led to the formation of new semantic associations in memory between a word and its newly acquired metaphorical meaning (e.g., the word “trench”, classically associated with the line of defense during the war, has turned into “hospital”, the place where the “war” against COVID-19 took place). Indeed, semantic memory is seen by many researchers as a network in which word concepts are nodes more or less interconnected to each other [34,35,36,37,38]. More specifically, word concepts can entertain taxonomic relationships within the network by sharing common semantic features (i.e., chair and bed), or they can be associated because they resort together or in the same context (i.e., car, petrol) [39]. Particularly, words that frequently co-occur in any language modality (spoken and/or written) become associated and, consequently, more interconnected in the semantic network [40,41]. Indeed, social context can influence our language use by making familiar words not previously frequently used [23,24]. Thus, given these premises, the hypothesis might be advanced that the overuse of newly acquired metaphors during the pandemic has reshaped our semantic memory, thus, establishing new semantic associations.

Since the pioneering study of Meyer and Schvaneveldt [42], theorists have used semantic priming tasks to investigate lexical access and semantic memory organization. The semantic priming effect has been mostly explored in lexical decision tasks in which participants are asked to decide whether prime-target pairs are or are not semantically related [43,44]. For instance, Meyer and Schvaneveldt [42] have shown that people are faster and more accurate in responding when the target name, such as “*nurse*”, is preceded by a semantically related prime, such as “*doctor*”, compared to an unrelated word such as “*butter*”. This facilitation has been consistently observed in several experiments and it occurs for word pairs that are semantically associated [36,45] (for a review, see [46]). The most common explanation related to this priming effect lies on the above hypothesis that semantically related words are represented closest to each other within the network. Thus, when a prime word is visually or auditorily presented, it automatically activates not only its own meaning but also the meaning of its semantically related targets decreasing the time to judge their appropriateness [34,35,47,48]. Many studies have shown that automatic effects predominate when the interval between the prime and the target is less than 400 msec (short stimulus onset asynchrony (SOA) [35], and that at longer SOAs strategic processes take over [49,50,51,52,53].

However, despite the huge amount of literature on semantic priming, to date, little attention has been given to explaining what conditions favor the formation of new associative links among words and, thus, consider these links as stored in semantic memory. A reasonable criterion is that the newly learned word pairs should exert the same priming effects as words already associated in semantic memory. Dagenbach and colleagues [54] investigated which learning conditions are essential to exert priming effects among pre-experimentally unrelated word pairs. They found that the frequent co-occurrence of these unrelated pairs during a study phase led to semantic priming effects approximately equal to those observed for pre-experimentally related word pairs. Based on these results, the authors concluded that these newly learned associations were added to semantic memory. Similar findings were reported in Schrijnemakers and Raaijmakers’s study [55], in which the presentation, for several trials, of newly learned word pairs led to faster reaction time in a lexical decision task compared to non-repeated word pairs associations.

Given these assumptions, in the present work, we wanted to investigate whether the presentation of word pairs acquired during the pandemic in which the target referred to the newly acquired metaphorical meaning of the prime (i.e., “ondata” [wave]—“contagio” [infection]) would exert the same semantic priming effect as word pairs with pre-consolidated semantic links (i.e., “ondata” [wave]—"maltempo” [bad weather]). We would expect that the overuse of these newly acquired word pairs has reshaped our semantic memory, thus, leading to the formation of new semantic associations.

## 2. Materials and Methods

### 2.1. Participants

One hundred and twenty bachelor’s degree students (60 female and 60 male) aged between 18 and 26 years old (mean = 21.4 years, SD = 2.40) were recruited for this study. Inclusion criteria were: Italian language as mother tongue; right-handed [56]; normal vision; and no history of chronic or acute neurologic, psychiatric, or medical disease. Among the study sample, no one was infected by COVID-19.

### 2.2. Ethics Statement

The data analyzed in the current study were collected in accordance with the Helsinki Declaration and the Institutional Review Board of the IRCCS Fondazione Santa Lucia, Rome, Italy. Prior to participation, all participants signed an online informed consent form.

### 2.3. Materials

In order to be sure that the relationship entertained between the prime and the target corresponded to a pre-existing semantic association (semantic pairs, i.e., “trincea” [trench]—“soldato” [soldier]), to a newly acquired association (pandemic pairs, i.e., “trincea” [trench]—“ospedale” [hospital]) or it did not correspond to any association (neutral pairs, i.e., “trincea” [trench]—“risposta” [response]), we first selected a sample of one-hundred-word pairs. Each pair of words belonging to the three categories had the same prime stimulus, while the target varied according to the type of relationship entertained with the prime. Sixty participants, matched for age and educational level to the experimental group (age mean = 21.73, SD = 2.35), who did not take part in the experiment, were asked to judge whether or not the relationship between the different word pairs corresponded to one of the three selected categories (semantic, pandemic, and neutral).

A final sample of sixty-word pairs, whose pairs were uniquely judged by all participants as belonging to the three categories, was finally selected. The sixty-word pairs were distributed into three lists, a list called “Semantic”, a “Pandemic” list, and a list called “Neutral”, each list was made of twenty-word pairs.

All targets belonging to the three lists were matched for frequency (list_pandemic_: mean = 39.25, SD = 63.94; list_semantic_: mean = 39.50, SD = 58.33; list_neutral_: mean = 37.75, SD = 56.52, unpaired *t*-test, *p* > 0.05 for each comparison) and length (list_pandemic_: mean = 7.1, SD = 1.65; list_semantic_: mean = 7.4, SD = 1.85; list_neutral_: mean = 7.2, SD = 1.96, unpaired t-test, *p* > 0.05 for each comparison) [57]. The targets were also matched for age of acquisition (list_pandemic_: mean = 8.76, SD = 3.14; list_semantic_: mean = 7.07, SD = 3.02; list_neutral_: mean = 7.57, SD = 3.02, unpaired *t*-test, *p* > 0.05 for each comparison) and imageability (list_pandemic_: mean = 5.07, SD = 0.94; list_semantic_: mean = 5.29, SD = 1.01; list_neutral_: mean = 5.58, SD = 1.22, unpaired *t*-test, *p* > 0.05 for each comparison) estimated on a sample of fifty participants (age mean = 21.65, SD = 2.58) along a 7-point Likert scale, respectively, for age of acquisition (from 1 = 0–2 years to 7 = 13+ years) and imageability (from 1 = no imageability to 7 = clear imageability) (adapted from [58]).

### 2.4. Procedure

The experiment was performed between 1st December 2021 and 15th February 2022. The lexical decision task was administered online through Psytoolkit software [59,60]. After reading the instructions and filling in a socio-demographic form, each participant was asked to digitally sign the consent form and then start the experiment. Each trial began with the presentation of a fixation cross in the center of the computer screen, which lasted 500 msec. After the extinction of the cross, the prime stimulus appeared for 800 msec in the same position as the cross. Then, the screen went blank for 1000 msec, and the target stimulus appeared for 800 msec. The screen went blank again for 1000 msec, and the next trial began. For each pair of stimuli, subjects had to decide whether the target was not related to the prime by pressing the L or the A button, respectively, on the computer’s keyboard as quickly as possible. Before starting the experiment, subjects were asked to train themselves by responding to ten related or unrelated word pairs. These pairs were used only for training purposes; thus, they did not appear in the experimental lists. The sixty pairs of stimuli belonging to the three lists were randomly presented across the experiment.

### 2.5. Data Analysis

The participant’s performance was evaluated by considering the mean percentage of response accuracy and the mean reaction times in milliseconds for each list of stimuli (pandemic vs. semantic vs. neutral). Data were analyzed with IBM SPSS Statistics 22 software. To verify the applicability of the parametric analysis, a Shapiro–Wilk normality test was used, which is the most powerful statistical instrument to measure the normality of the data [61]. The test revealed a normal distribution of the data. Statistical analyses were performed through two separate repeated measures ANOVAs, respectively, for response accuracy and reaction times (RTs). For each analysis, CONDITION (pandemic vs. semantic vs. neutral) was considered as the only “within” factor. If the ANOVAs showed significant effects, respective post-hoc Bonferroni tests were conducted. We used the Bonferroni multiple comparisons method because it is the most conservative test to investigate statistical significance among the different comparisons [62].

## 3. Results

### 3.1. Accuracy

The analysis showed a significant effect of CONDITION (pandemic vs. semantic vs. neutral, F (2,238) = 21.07, *p* < *0*.001). The Bonferroni’s post-hoc test revealed that the mean percentage of accuracy was significantly greater for the pandemic than for the semantic (pandemic 94% vs. semantic 93%, *p* = 0.001) and the neutral pairs (pandemic 94% vs. neutral 92%, *p* < 0.001). The mean percentage of accuracy was also greater for the semantic than the neutral pairs (semantic 93% vs. neutral 92%, *p* = 0.02) (see Figure 1).

### 3.2. Reaction Times (RTs)

The analysis showed a significant effect of CONDITION (pandemic vs. semantic vs. neutral, F (2,238) = 161.43, *p* < 0.001). The Bonferroni’s post-hoc test revealed faster RTs in the pandemic compared to the semantic pairs (pandemic 1065.14 msec vs. semantic 1216.76 msec, *p* < 0.001) and in the semantic compared to the neutral pairs (semantic 1216.76 msec vs. neutral 1299.60 msec, *p* < 0.001). RTs were also significantly faster between the pandemic and the neutral pairs (pandemic 1065.14 msec vs. neutral 1299.60 msec, *p* < 0.001) (see Figure 2).

## 4. Discussion

The aim of the present study was to investigate whether the frequent co-occurrence of word pairs during the pandemic would lead to the formation of new semantic associations in memory. To verify this, in a lexical decision task, participants’ performance on pandemic word pairs, in which the target referred to the newly acquired metaphorical meaning of the prime (i.e., “trench”—“hospital”), as compared to their responses to word pairs with pre-established semantic links (i.e., “trench”—“soldier”) and to neutral pairs (i.e., “trench”—“response”). Results revealed greater accuracy and faster response times for pandemic pairs than for semantic ones. Thus, the repeated co-occurrence of newly learned words has led to semantic priming effects stronger than those observed for words with pre-consolidated semantic links suggesting the formation of the new semantic association in memory. More accurate and faster responses were also present for the pandemic pairs with respect to the neutral pairs. This result, thus, validates the hypothesis that these newly acquired associations depended on the co-occurrence of those word pairs during the pandemic and not on possible occasional associations between different words. The results also revealed significant differences in accuracy and in response times between the semantic pairs compared to the neutral ones. This latter result confirms previous findings in semantic priming studies using lexical decision tasks, which suggested that semantically associated word pairs are more likely to exert faster responses than neutral pairs [42]. Indeed, the cognitive representation of words is characterized by two dimensions. Their frequency effect depends on how often and in what contexts they appear and their semantic characteristics. This latter gives rise to the well-known semantic priming effect in which the presentation of a word (i.e., dog) facilitates the recognition of an associatively related, subsequently presented word (i.e., cat) [34,35,36,37,42]. From the beginning of the pandemic, the people’s mental lexicon underwent a drastic change, influencing both dimensions. Uncommon words such as “trench” increased in their frequency of use for millions of people, while, in parallel, words with minimal pre-existing associations, such as “wave” and “infection”, became closely linked nodes stored in semantic memory. Thus, we believe that the continuous and persistent exposure to pandemic pairs has somehow reshaped our semantic memory making the recognition of these word pairs easier than the semantic and neutral ones.

Our results are in line with other evidence showing that repeated exposure to new words facilitates the learning and merging of these words in the adult lexicon [55,63]. Indeed, Coane and Balota [64] have suggested that semantic priming effects might be supported by two general classes of semantic models. Feature-overlap models assume that shared semantic features between primes and targets are critical (e.g., cat-DOG), and associative models affirm that contextual co-occurrence is critical and that the system is organized along associations independent of featural overlap (e.g., leash-DOG). Thus, if previously unrelated concepts become related as a result of their contextual co-occurrence, as during the pandemic, this would create new associative links among them, which, in turn, result in semantic priming effects.

In line with our data, recent literature has suggested that, as a consequence of the pandemic, specialized domain elements have been transferred to the common language. Indeed, very recently, Birò et al. [65] pointed out that while, during the pandemic, some words have assumed a more restricted meaning, a grammatical principle known as “specialization” [66], others were referred to a more general meaning, the opposite phenomenon known as “de-categorization” [66]. An illustration of the first principle can be found in the use of the word “quarantine” or “positive”; the latter principle, which was more diffused, has influenced terms such as “super-spreader” or “FFP2 mask”. Thus, during the pandemic, the repetitive association of a word with a newly acquired metaphorical meaning has weakened its original meaning, making its figurative meaning predominant (e.g., the word “trench”, classically associated with the line of defense during the war, has turned into its figurative meaning of “hospital”, the place where the “war” against COVID-19 took place). Indeed, the peculiarity of this health crisis was the overuse of military metaphors evoking images of conflict, enemies, and death which dramatically increased the levels of anxiety, depression, and stress among people worldwide [67,68]. Accordingly, very recently, Georgiu [69] investigated whether exposure to media expressions containing alarming and militaristic language (i.e., war) affected people’s feelings with respect to the pandemic. Results showed that individuals who were exposed to these expressions were more pessimistic in judging the impact of the virus on their health than those who were exposed to more neutral language [69]. Similarly, in Panchuelo et al.’s work [70], significant changes were reported in emotional arousal for recurrent COVID-19-related words (i.e., hospital) with respect to unrelated words (i.e., whale), thus, suggesting that the pandemic context has modified the affective representation of its related words [70]. Indeed, since emotional processes are an active function of social events [71], these findings support, in line with our results, the flexibility of emotional representations and the malleability and dynamicity of our mental lexicon as a function of contextual factors.

## 5. Conclusions

We believe that our work highlights two important findings with opposite connotations. On one side, for the first time, it shows that the frequent co-occurrence of words during the pandemic has created new associative links in our semantic memory. Thus, our results confirm the dynamic property of human language and how it can be influenced by the social context to which it belongs. On the other, it underlines how these changes in language use could negatively impact the social context itself. Indeed, during the pandemic, the overuse of metaphors evoking dramatic images has contributed to the increase of several psychological disorders which are still reported worldwide nowadays. In this context, further research is urgently needed, first of all, to investigate whether the results obtained can be generalized to other populations and to establish whether these newly learned associations will still be consolidated in our memory as the pandemic is definitively defeated.

## Figures and Tables

**Figure 1 ijerph-19-13836-f001:**
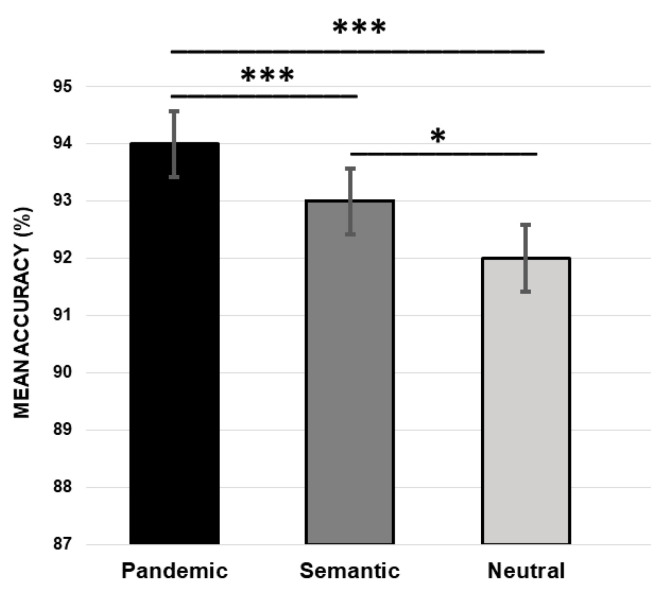
Mean percentage of response accuracy for pandemic, semantic, and neutral stimuli (*** *p* ≤ 0.001, * *p* ≤ 0.05).

**Figure 2 ijerph-19-13836-f002:**
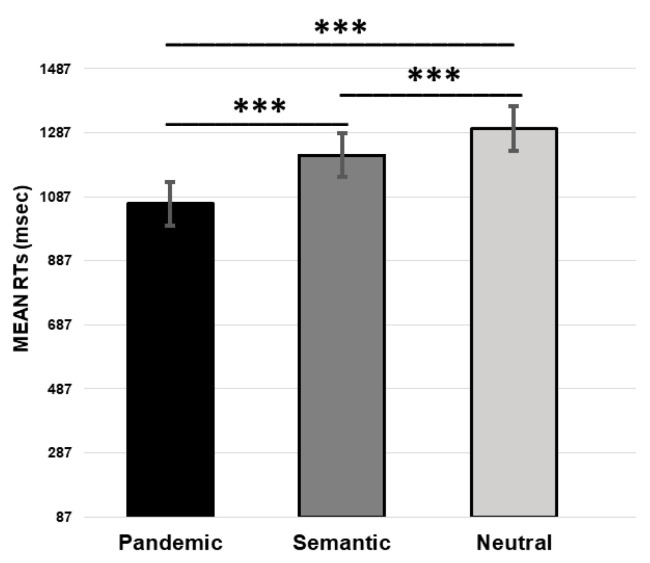
Mean percentage of reaction times (RTs) for pandemic, semantic, and neutral stimuli (*** *p* ≤ 0.001).

## Data Availability

The data presented in this study are available on request from the corresponding author. The data are not publicly available due to ethical and privacy restrictions.

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
