# Peer review of "How Has COVID-19 Impacted Our Language Use?"

_ijerph, 2022, doi:10.3390/ijerph192113836_

Round 1
Reviewer 1 Report
1. Why did the authors select all the participants are bachelors? Is it possible that participants have various levels of education? Moreover, what is the reason that the participants should be right-handed?
2. The authors should describe what is the content of the preliminary sample of one hundred pairs and the reason why they were adopted in their study. Moreover, the authors also should explain how they select the sixty pairs from the preliminary sample of one hundred pairs.
3. What is the hypothesis of this study? The authors should the study hypothesis in the manuscript.
4. Abbreviations that appear first in the manuscript should have their full text. Moreover, the authors should use the same abbreviation to express the standard deviation (Please see line 149).
5. The authors should have more descriptions to explain how they got the mean values and standard deviations of the frequency, length, and age of acquisition shown in the last paragraph of page 3.
6. The authors used the Shapiro–Wilk normality test to verify the applicability of the parametric analysis in their study; they should have some descriptions to explain the Shapiro–Wilk normality test in the manuscript. Moreover, the authors also used Bonferroni’s post-hoc test to reveal the mean percentage of accuracy and reaction time; they should have some descriptions to explain Bonferroni’s post-hoc test in the manuscript.
7. Examining Figure 1 and Figure 2, Figure 2 shows the three comparison results in reaction time among pandemic, semantic and neutral stimuli; however, Figure 1 only shows the two comparison results in response accuracy among pandemic, semantic and neutral stimuli. Why does not Figure 1 shows three comparison results in response accuracy among pandemic, semantic and neutral stimuli?
8. The authors should further explore the reasons why the COVID-19 impacted on our language use in the discussion section.
Author Response
- Why did the authors select all the participants are bachelors? Is it possible that participants have various levels of education? Moreover, what is the reason that the participants should be right-handed?
We wanted the group to be homogeneous with the same level of education, age, and handedness. As you know, in left-handed subjects the representation of language can be different from right-handed subjects. For this reason, we have only chosen right-handed participants. Surely, it would have been interesting to investigate the impact of Covid-19 on language use also in different groups of subjects but, during the pandemic, as you know, with the exception of students who attended university’s courses, it was very difficult to recruit people willing to participate in experiments. We have highlighted this last point in the conclusions section.
- The authors should describe what is the content of the preliminary sample of one hundred pairs and the reason why they were adopted in their study. Moreover, the authors also should explain how they select the sixty pairs from the preliminary sample of one hundred pairs.
We really thank the Referee for his/her request of clarification. In order to be sure that the relationship entertained between the prime and the target corresponded to a pre-existing semantic association (semantic pairs, i.e., ‘trincea’ [trench] – ‘soldato’ [soldier]), to a newly acquired association (pandemic pairs, i.e., ‘trincea’ [trench] – ‘ospedale’ [hospital]) or it did not correspond to any association (neutral pairs, i.e., ‘trincea’ [trench] – ‘risposta’ [response]), we first selected a sample of one hundred-word pairs. Each pair of words, belonging to the three categories, had the same prime stimulus while the target varied according to the type of relationship entertained with the prime. Sixty participants, matched for age and educational level to the experimental group (age mean=21.73, SD=2.35), who did not take part in the experiment, were asked to judge whether or not the relationship between the different word pairs corresponded to one of the three selected categories (semantic, pandemic, and neutral). A final sample of sixty-word pairs, whose pairs were uniquely judged by all participants as belonging to the three categories, was finally selected. The sixty-word pairs were distributed into three lists, a list called “Semantic”, a “Pandemic” list and a list called “Neutral”, each list made of twenty-word pairs. As requested by the Referee, in the revised version, we have clarified the materials section.
- What is the hypothesis of this study? The authors should the study hypothesis in the manuscript.
We agree with the Referee’s suggestion that our study’s hypothesis should have been better explained. In the revised version, we have clarified this point.
- Abbreviations that appear first in the manuscript should have their full text. Moreover, the authors should use the same abbreviation to express the standard deviation (Please see line 149).
We are sorry for these omissions. In the revised version, we have corrected our oversights.
- The authors should have more descriptions to explain how they got the mean values and standard deviations of the frequency, length, and age of acquisition shown in the last paragraph of page 3.
We agree with the Referee’s suggestion. Accordingly, as above written, we have revised the materials section. To calculate the mean frequency and length and SD of our word pairs sample, we referred to a corpus and lexicon of frequency and length of written Italian words which can be consulted free of charge on the web (please see reference [60]). To calculate the mean values and SD of age of acquisition and imageability, we used a 7-point Likert scale, respectively, for age of acquisition (from 1= 0-2 years to 7= 13+ years) and imageability (from 1=no imageability to 7=clear imageability) (please see reference [61]).
- The authors used the Shapiro–Wilk normality test to verify the applicability of the parametric analysis in their study; they should have some descriptions to explain the Shapiro–Wilk normality test in the manuscript. Moreover, the authors also used Bonferroni’s post-hoc test to reveal the mean percentage of accuracy and reaction time; they should have some descriptions to explain Bonferroni’s post-hoc test in the manuscript.
We thank the Referee for his/her suggestion. Accordingly, in the Data analysis section, we briefly explained why e used the Shapiro test and the Bonferroni post-hoc test.
- Examining Figure 1 and Figure 2, Figure 2 shows the three comparison results in reaction time among pandemic, semantic and neutral stimuli; however, Figure 1 only shows the two comparison results in response accuracy among pandemic, semantic and neutral stimuli. Why does not Figure 1 shows three comparison results in response accuracy among pandemic, semantic and neutral stimuli?
We agree with the Referee’s request, and we are sorry for this omission. Accordingly, we revised Figure 1 also adding the three comparisons.
- The authors should further explore the reasons why the COVID-19 impacted on our language use in the discussion section.
We really thank the Referee for this request. As the Referee can see, in the revised version, in different paragraphs, we have expanded our discussion and explanation on the impact of Covid-19 on our language use.
Reviewer 2 Report
The question was if the excessive use of metaphors of semantic terms related to COVID-19 and the pandemic have led to the formation of new semantic associations in people's memory, to do so, it analyzes how pairs of words learned in the pandemic had a semantic effect than pairs of words already consolidated prior to the pandemic.changed people at all levels and it is interesting to note that also in the concept and association of words that have been used very frequently. It is original, as far as it is relevant, it is a bit holistic and comprehensive of the changes that the pandemic has brought about in society, this may be little perceived by everyone and by reading the article you appreciate how it is a change that also exists.
The assessment of language as an agent that affects us psychologically, how words influence mental health and how the use of language can change society over time. Since these terms are here to stay, it is important to analyze how they are now conceived. From my point of view, it is a novel appreciation that has not been previously published.
They could include people in all age ranges and not just university students in the analysis, since this is biased and the results cannot be generalized.
The question was whether the excessive use of metaphors of semantic terms related to COVID-19 and the pandemic have led to the formation of new semantic associations in people's memory, and in the conclusions it begins by responding to this, with the affirmation of that the results confirm the dynamic property of human language and how it can be influenced by the social context to which it belongs. And that the excessive use of metaphors that evoke dramatic imagery has contributed to the rise of various psychological disorders that are still reported around the world today.
In relation to the topic to be discussed, although the number may be excessive, since the introduction already includes 55 citations
The two figures they provide are not very representative, they do not provide clarity or informative value.
Author Response
- From my point of view, it is a novel appreciation that has not been previously published. They could include people in all age ranges and not just university students in the analysis, since this is biased, and the results cannot be generalized.
We really thank the Referee for his/her positive comments and we agree with the Referee’s criticism. Surely, it would have been interesting to investigate the impact of Covid-19 on language use also in groups of subjects different from university’s students. However, during the pandemic, as the Referee certainly knows, with the exception of students who attended university’s courses, it was very difficult to recruit people willing to participate in experiments. Following the Referee’s observation, in the conclusion section, we have highlighted that further research is needed in this field to investigate if the results obtained can be generalized to different populations.
- The two figures they provide are not very representative, they do not provide clarity or informative value.
We agree with the Referee’s suggestion. Accordingly, we have revised both figures.
Round 2
Reviewer 1 Report
The authors depended on the comments to revise the manuscript.